# Two critical temperatures conundrum in La$_{1.83}$Sr$_{0.17}$CuO$_4$

Abhisek Samanta[1], Itay Mangel[2], Amit Keren[2], Daniel P. Arovas[3], Assa Auerbach[2]

**1** Department of Physics, The Ohio State University, Columbus OH 43210, USA,
**2** Physics Department, Technion, Haifa, Israel
**3** Department of Physics, University of California at San Diego,
La Jolla, California 92093, USA
*Corresponding: assa@physics.technion.ac.il

May 3, 2024

## Abstract

**The in-plane and out-of-plane superconducting stiffness of La$_{1.83}$Sr$_{0.17}$CuO$_4$ rings appear to vanish at different transition temperatures, which contradicts thermodynamical expectation. In addition, we observe a surprisingly strong dependence of the out-of-plane stiffness transition on sample width. With evidence from Monte Carlo simulations, this effect is explained by very small ratio $\alpha$ of inter-plane over intra-plane Josephson couplings. For three dimensional rings of millimeter dimensions, a crossover from layered three dimensional to quasi one dimensional behavior occurs at temperatures near the thermodynamic transition temperature $T_c$, and the out-of-plane stiffness *appears* to vanish below $T_c$ by a temperature shift of order $\alpha L_a/\xi^{\parallel}$, where $L_a/\xi^{\parallel}$ is the sample's width over coherence length. Including the effects of layer-correlated disorder, the measured temperature shifts can be fit by a value of $\alpha = 4.1 \times 10^{-5}$, near $T_c$, which is significantly lower than its previously measured value near zero temperature.**

## 1 Introduction

A homogeneous three-dimensional superconductor is expected to exhibit a single transition temperature $T_c$ at which the order parameter, $\Delta(T)$, and all the superconducting stiffness components vanish [1,2]. In this regard, recent measurements of the *ab*-plane ($\rho^{\parallel}$) and *c*-axis ($\rho^{\perp}$) stiffnesses of La$_{1.875}$Sr$_{0.125}$CuO$_4$ crystals by Kapon *et. al.* [3] have been puzzling. Counter to the expectation above, $\rho^{\perp}$ was seen to vanish at $T_c^{\perp}$, which is about $0.64\,\mathrm{K}$ below the vanishing temperature $T_c^{\parallel}$ of $\rho^{\parallel}$.

*Disorder* – Short range uncorrelated disorder is not expected to affect the critical behavior of a superconductor, by Harris's criterion [4]. On the other hand, the cuprates are known to be highly anisotropic layered superconductors. Layer-correlated disorder, (or a gradient in dopant concentration along the *c* axis) [5,6], yields a distribution of $\rho^{\parallel}$ and $T_c^{\parallel}$. Experimentally, such inhomogeneity is manifested by a high temperature tail of the measured $\rho^{\parallel}$ above the average $T_c^{\parallel}$, while $\rho^{\perp}$ vanishes at the lowest values of $T_c^{\parallel}$ (see Appendix A). However, $T_c^{\parallel}$ in Ref. [3] exhibited inhomogeneity broadening of $\sim 0.1\,\mathrm{K}$, which is significantly below the apparent difference in $T_c$'s.

*Finite size effects* – An alternative proposition is that finite sample dimensions play a role. Previous Monte-Carlo simulations [7,8] of the 3dXY model found strong effects of sample dimensions on the temperature dependent stiffness coefficients. These effects are expected to be enhanced by high anisotropy.

This paper explores finite size effects experimentally and theoretically. We report systematic stiffness measurements near $T_c$ for $La_{1.83}Sr_{0.17}CuO_4$ rings with widths $L_a, L_c$ ranging between $L = 0.1$ to 1 millimeter. $T_c^{\parallel}$ is found to be weakly dependent on $L_c, L_a$. In contrast, a significant reduction of $T_c^{\perp}$ for decreasing width $L_a$ is observed. This behavior is not expected for layer-correlated inhomogeneity. The relatively strong finite size effect demands theoretical explanation.

Phenomenologically, the monotonous relation between $T_c$ and $\rho^{\parallel}$ in cuprates [9], and the observed jump in $\rho^{\parallel}$ at $T_c$ in ultra-thin films [10], suggest that $T_c$ is driven by super-conducting phase fluctuations [11], and vortex unbinding [12]. Therefore we appeal to the three dimensional classical XY (3dXY) model (rather than BCS theory) to explain the stiffness temperature dependence toward $T_c$.

We applied a Monte-Carlo simulation with Wolff cluster updates on finite three dimensional lattices. The in-plane and intra-plane superconducting stiffness coefficients of the highly anisotropic 3dXY model appeared to vanish at different transition temperatures. The numerical simulations showed a strong dependence of the apparent inter-plane stiffness vanishing temperature $T_c^{\perp}$ on the layers' finite width. This dependence exceeded the magnitude expected of critical fluctuations.

The numerical and experimental observations are understood as follows. Inter-layer mean field theory [13], predicts a thermodynamic transition temperature slightly above the two dimensional Berezinskii-Kosterlitz-Thouless [12] (BKT) transition at $T_{\mathrm{BKT}} < T_c$. 3dXY critical behavior [1] is expected be observed only very close to $T_c$. As temperature approaches $T_c$, the finite sample width $L_a$ drives a crossover of $\rho^{\perp}$ to the stiffness of a one dimensional $XY$ (1dXY) chain [8]. This crossover results in an exponentially flat temperature dependence of $\rho^{\perp}(T, L_a)$ below $T_c$. For finite experimental or numerical resolution, such singular behavior always appears as vanishing of $\rho^{\perp}$ at $T_c^{\perp}(L_a) < T_c$.

We compare our theoretical analysis to the experimental values $T_c^{\perp}(L_a)$, and use the fit to estimate of the anisotropy parameter of $La_{1.83}Sr_{0.17}CuO_4$ near its thermodynamic $T_c$.

## 2 Experiments

Measurements were carried out with a 'stiffnessometer' apparatus [14] which comprises of a long excitation coil piercing a superconducting ring. A bias current in the coil creates an Aharonov-Bohm (AB) vector potential $\boldsymbol{A}$ which, by London's equation, produces a persistent current that is measured by the induced (dia-)magnetization $m^{\alpha}$ along the coil axis $\alpha$. One then measures $m^{\alpha}$ by moving a pickup loop relative to the ring and coil. The apparatus is shown in Fig. 1.

$La_{2-x}Sr_xCuO_4$ is known to grow in large single crystals allowing significant size reductions. Therefore, powder of different doping is prepared from stoichiometric ratios of $99.99\%$ pure $CuO$, $La_2O_3$, and $SrCO_3$ to make feed and seed rods. This powder is turned into a single crystal using an image furnace with four elliptic mirrors focusing 300 W halogen lamps. The growth was stabilised over 100 hr without any change of the lamp 59% power. Growth rate of 1.0 mm/h, down-ward translation of 0.15 mm/h, and rotation in opposite directions at 15 rmp were used. The emerging crystals looked like Fig. 1 of Ref. [15]. After the growth, the crystals were annealed in argon environment at $T = 850$ C for 120 hr to release internal stress. Finally, the crystals were oriented with a Laue camera, and cut into rings with a femtosecond laser cutter. For each doping two rings, labelled by $a$ and $c$, were prepared with their coil axes parallel $(a)$ and perpendicular $(c)$ to the $CuO_2$ planes. The width and height of the rings was varied by the laser, or polished down to

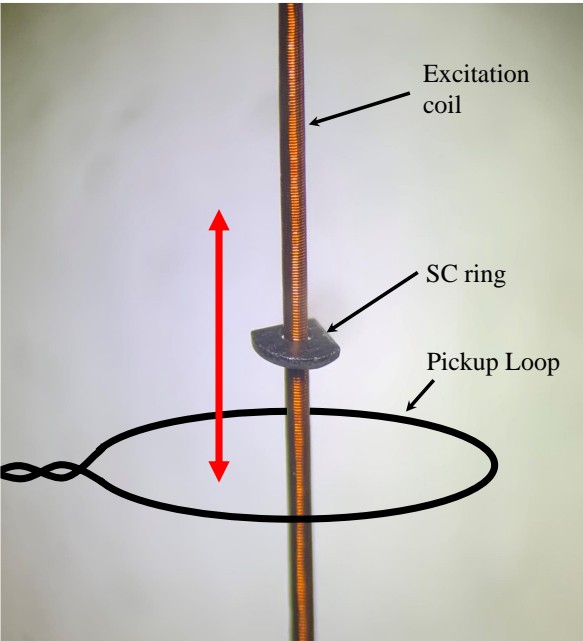

Figure 1: A superconducting ring cut in two directions, on the excitation coil. The red double arrow shows the moving direction of the schematic magnetization measuring pickup-loop, relative to both coil and ring.

the geometries shown in Fig. 2a,b.

For the $a$-ring, we varied mostly the narrowest (bottleneck) widths of the $a-b$ planes, $L_a$, whereas for the $c$-ring we varied both $L_c$ and $L_a$ (see Figs. 2a,b). Fig. 2(c) shows the narrowest bottleneck geometry of the $a$-ring. The requirement to: cut, measure, cut, measure, et cetera, the same pair of samples proved challenging. In most cases one of the samples broke during some step of the process. Only one pair of $La_{1.83}Sr_{0.17}CuO_4$ rings survived the reduction of $L_a$ by factor of 10 between the initial and final cutting stages. The magnetization of this sample is depicted in Fig. 2.

When the transverse London penetration depth $\lambda_c$ ($\lambda_a$) is smaller than the sample width $L_c$ ($L_a$), the induced persistent current in the superconductor precisely cancels the AB flux of the coil. This results in a temperature independent induced magnetization $m^a$ ($m^c$) at low temperatures.

As $T \to T_c$, the AB flux in the coil is under screened as $\lambda_\alpha(T) \geq L_\alpha$. In this temperature regime $m^\alpha(T)$ decreases rapidly and becomes linearly proportional to the in-plane stiffness components. As an example, for the geometry of a perfect ring

$$
m^{a,c}(T) = h \int_{r_{\rm in}}^{r_{\rm out}} dr \, \pi r^2 \, (\hat{\boldsymbol{r}} \times \boldsymbol{j}_{\rm sc}(r))^{a,c}
$$
$$
= -h(r_{\rm out}^2 - r_{\rm in}^2) \, \frac{\Phi}{4} \, \rho^{\perp,\parallel}(T) \quad . \tag{1}
$$

$h$, $r_{\rm in}$, and $r_{\rm out}$ are the ring's height, and inner and outer radii, and $\Phi$ is the flux produced by the coil. For irregular rings extracting $\rho^\perp, \rho^\parallel$ from $m^\alpha(T)$ requires a full solution of Ginzburg-Landau and Biot-Savart equations [16]. However, here we do not require the magnitude of $\rho^\perp, \rho^\parallel$ but only their vanishing temperatures $T_c^\perp$ and $T_c^\parallel$. These are experimentally determined by the vanishing of the corresponding magnetizations.

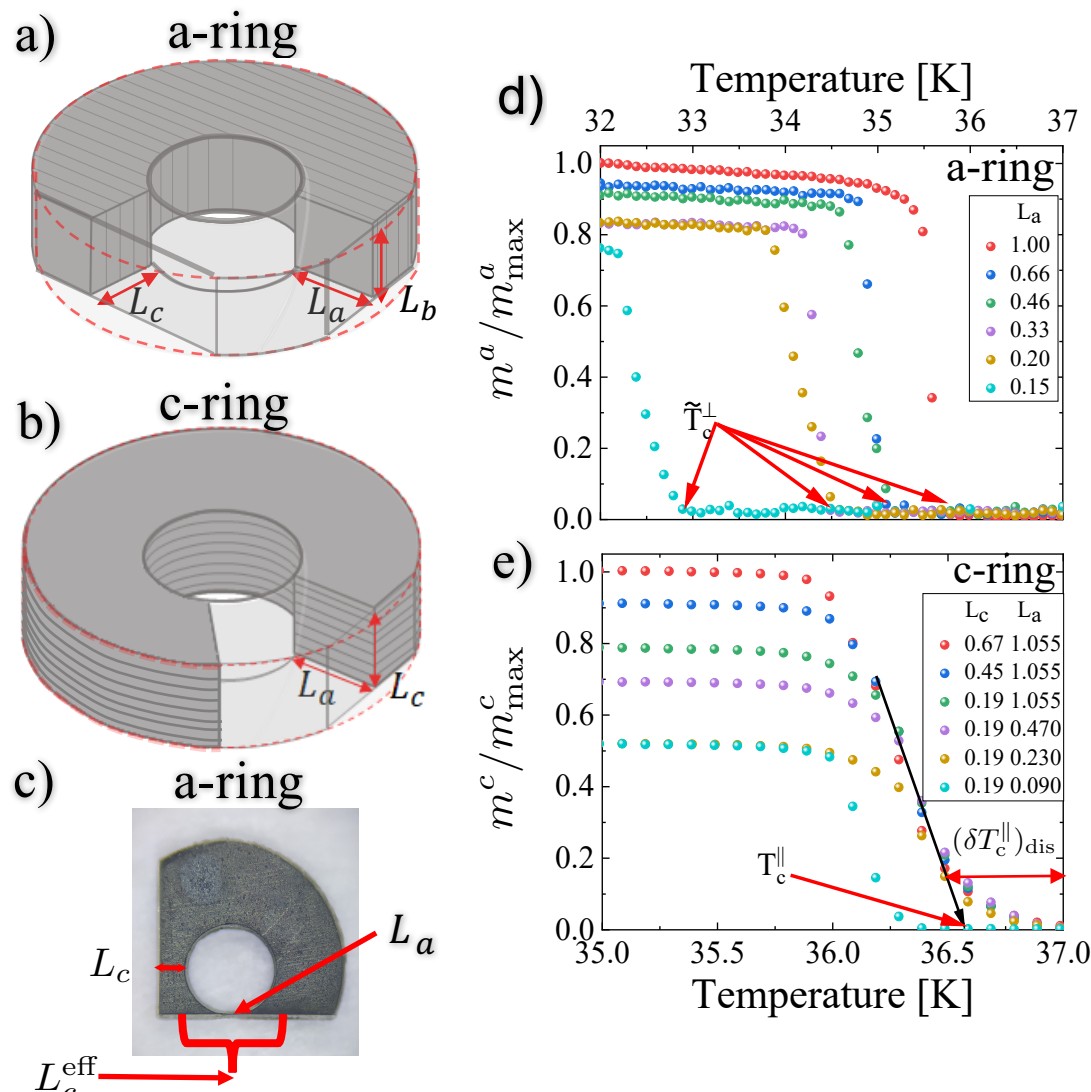

Figure 2: Experimental configuration and normalized magnetizations of $La_{1.83}Sr_{0.17}CuO_4$ rings for a fixed 1 mA current in the coil. (a) The interior of the $a$-ring. The $CuO_2$ planes are parallel to the ring's symmetry axis. This ring is sequentially polished to reduce the layers' width $L_a$ in the bottleneck region. (b) The interior of the $c$-ring. The $CuO_2$ planes are perpendicular to the ring's symmetry axis. This ring is polished along two planes which varies both $L_a$ and $L_c$. (c) Photograph of an $a$-ring with two cut planes. $L_c^{\text{eff}}$ defines the effective aspect ratio in the bottleneck region. (d) Magnetization $m^a$ of $a$-rings with variable $L_a$. The apparent stiffness vanishing temperatures are denoted by $\widetilde{T}_c^\perp(L_a)$. (e) Magnetization $m^c$ of $c$-rings. Except for the narrowest sample $L_a = 0.09$ (which is suspected of containing a traversing cut), the magnetizations near their transition are insensitive to $L_a$. The tail of width $(\delta T_c^\parallel)_{\text{dis}} \simeq 0.5\,\text{K}$ is assumed to reflect some layer-correlated disorder, which is a smaller effect than the finite size dependence of the $a$-rings' $\widetilde{T}_c^\perp$. $\bar{T}_c^\parallel$ is averaged in-plane transition temperature (see Section 6 and Appendix A).

Figs. 2(d,e) depict the temperature-dependent relative magnetizations $m^\alpha(T, L_a)/m_{\text{max}}^\alpha$, for $\alpha = a, c$. $m_{\text{max}}^\alpha$ is the zero temperature magnetization of the largest ring. Fig. 2(d) shows a strong dependence of the $a$-ring's magnetization apparent vanishing temperature

$\tilde{T}_c^\perp$ on the transverse width $L_a$. In contrast, the $c$-rings' magnetization in Fig. 2(e), exhibit insensitivity to the sample widths in the ranges $L_a \in [1.05, 0.23]$ and $L_c \in [0.67, 0.19]$. We note an exception of the $(L_a, L_c) = (0.09, 019)$ mm sample, which we believe to be damaged by a deep fracture during the cutting process.

We note that the $c$-ring magnetizations exhibit a high temperature tail of $\simeq 0.5\,\mathrm{K}$ above the extrapolated transition at $\bar{T}_c^\parallel$. This is attributed to layer-correlated inhomogeneity as discussed in the Introduction and Appendix A. This inhomogeneous broadening will be taken into account in fitting theory to the experimental data in Section 3.

## 3   Layered 3dXY model

As mentioned before, we model the phase fluctuations of $\mathrm{La}_{1.83}\mathrm{Sr}_{0.17}\mathrm{CuO}_4$ near $T_c$ by the classical 3dXY Hamiltonian on a tetragonal lattice,

$$H_{3dXY} = -\sum_i \sum_\gamma J_\gamma \cos(\varphi_{\boldsymbol{r}_i} - \varphi_{\boldsymbol{r}_i + \boldsymbol{a}_\gamma}) \quad , \tag{2}$$

where $\gamma \in \{a, b, c\}$ and where $J_a = J_b = J^\parallel$ and $J_c = J^\perp$ are the effective intra- and inter-plane Josephson couplings. The effective anisotropy parameter is defined as $\alpha = J^\perp / J^\parallel$. $\alpha$ will later be determined to fit experimental data near $T_c$. The two dimensional limit $\alpha = 0$ reduces to the two dimensional $XY$ (2DXY) model, where by Mermin and Wagner theorem the superconducting order parameter $\Delta = \langle e^{i\varphi} \rangle$, and $\rho^\perp$ vanish at all temperatures. Nevertheless, the in-plane stiffness is non-zero below $T_{\mathrm{BKT}} \simeq 0.893 J_a$.

For small but finite anisotropy $0 < \alpha \ll 1$, inter-layer mean field theory (IMFT) is very useful [13, 17–20]. It predicts $\Delta(T) > 0$ for $T < T_c$, where $T_c$ is the three dimensional critical temperature. IMFT uses the exponential divergence of the BKT susceptibility above $T_{\mathrm{BKT}}$ to obtain,

$$\frac{T_c(\alpha) - T_{\mathrm{BKT}}}{T_{\mathrm{BKT}}} = \left( \frac{b}{|\ln(0.14\alpha)|} \right)^2 \quad . \tag{3}$$

Here, the (non universal) constant is taken to be $b = 2.725$ [21].

In the regime $[0, T_{\mathrm{BKT}}]$, the order parameter magnitude $\Delta = |\langle e^{i\varphi} \rangle|$ decreases from unity as calculated by Hikami and Tsuneto [22],

$$\Delta^2(T)_{T \leq T_{\mathrm{BKT}}} \simeq \exp\left( -\frac{T \log(1/\alpha)}{4\pi J_a} \right) \quad . \tag{4}$$

Over the crossover regime $T \in [T_{\mathrm{BKT}}, T_c]$, the order parameter squared initially crosses over with an intermediate power law of $|T - T^*|^{0.46}$, where $T^* = T_{\mathrm{BKT}} + \frac{1}{4}(T_c - T_{\mathrm{BKT}})$ [19], above which it drops precipitously toward $T_c$ as,

$$\Delta^2(T) = \Delta_{\mathrm{BKT}}^2 \, t^{2\beta} \quad , \quad t \equiv \left( \frac{T_c - T}{T_c - T_{\mathrm{BKT}}} \right) \quad , \tag{5}$$

where $\beta$ crosses over from the mean field value $\frac{1}{2}$ to the 3dXY exponent 0.349, within a narrow three dimensional Ginzburg critical region of width $T_c / \log^4(\alpha)$ [13].

Fig. 3 depicts the smoothed "trapezoidal" temperature dependence of $\Delta^2$ which differs from the BCS theory for the gap squared. We note that the spectral gaps observed by photoemission do not directly measure the thermodynamic order parameter. In the underdoped pseudogap phase [23], parts of the Fermi surface gap survives above $T_c$ [24, 25]. A more direct measurement of $\Delta^2$ near $T_c$ would be the superconducting stiffness [1, 13], since

$$\rho_\gamma(T) \propto \Delta^{2 - \eta\nu\beta^{-1}} \tag{6}$$

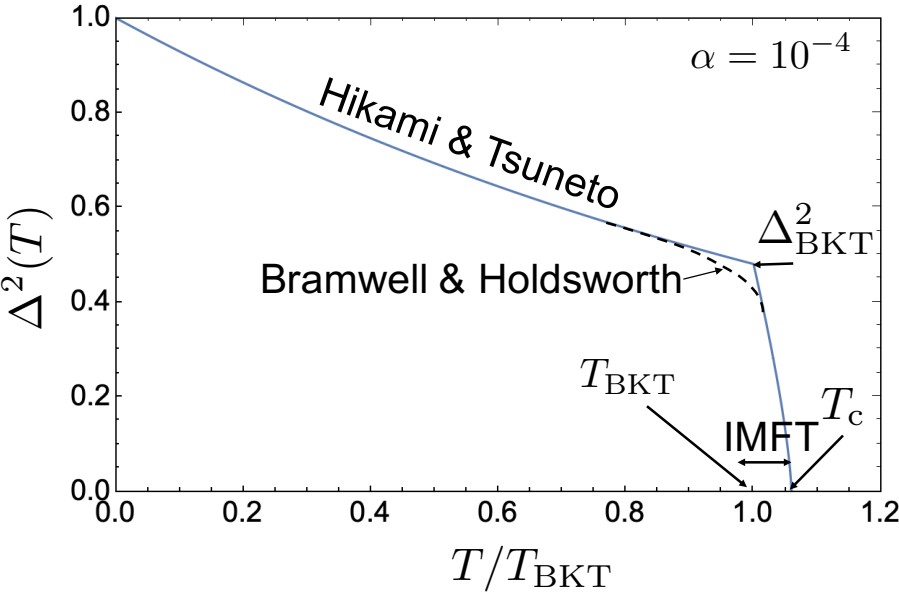

Figure 3: The order parameter squared as a function of temperature for the layered classical XY model, for anisotropy parameter $\alpha = 10^{-4}$. The graph patches the linear spinwave theory of Hikami and Tsuneto [22], the crossover (dashed line) power law of Bramwell and Holdsworth [19], and the three dimensional critical point which is obtained by Inter-plane Mean Field Theory (IMFT) of Eqs. (3), (4) and (5).

where $\eta$ and $\nu$ are the critical correlation function power law and correlation length exponents respectively. For the 3dXY model $\eta\nu = 0.0255$ which is small and henceforth neglected.

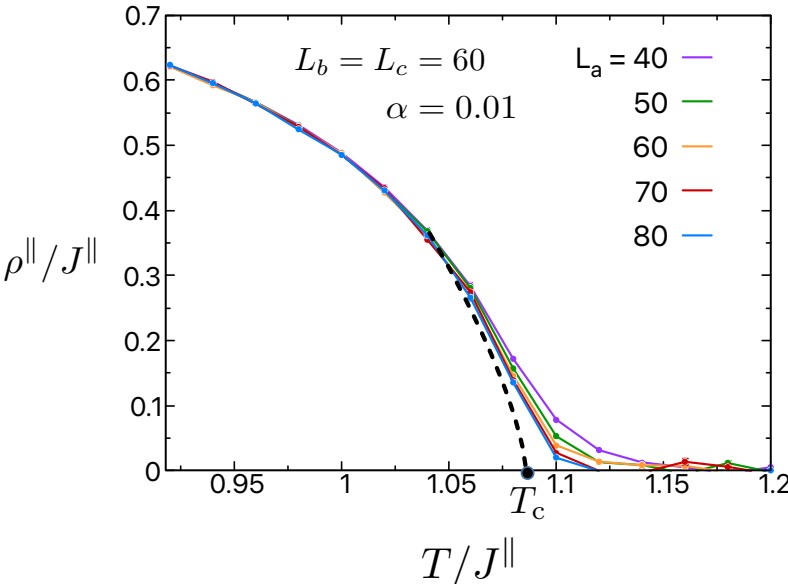

Figure 4: The intra-plane stiffness $\rho^{\parallel}$, plotted as a function of temperature $T$, for $\alpha = 0.01$ and for different $L_a$ between 40 and 80, while $L_b$ and $L_c$ are kept fixed at 60. The error bars are smaller than the point sizes. The black dashed line shows the critical behavior near the thermodynamic transition temperature $T_c$, according to Eq. (6).

## 4  Monte Carlo simulations

The superfluid stiffness (*i.e.* helicity modulus) of Eq. 2 with $a_\gamma = 1$, is given by [7, 26]

$$
\rho_\gamma = \frac{J_\gamma}{V} \left\langle \sum_{\langle ij \rangle} \cos(\varphi_{\boldsymbol{r}_i} - \varphi_{\boldsymbol{r}_j}) \, (r_i^\gamma - r_j^\gamma)^2 \right\rangle
$$
$$
- \frac{J_\gamma^2}{VT} \left\langle \left( \sum_{\langle ij \rangle} \sin(\varphi_{\boldsymbol{r}_i} - \varphi_{\boldsymbol{r}_j}) \, (r_i^\gamma - r_j^\gamma) \right)^2 \right\rangle \quad , \gamma = a, b, c.
$$
(7)

$V = L_a L_b L_c$. The first contribution measures the short range correlations, which are proportional to minus the energy along the bonds in the $\gamma$ direction. The second contribution measures long range current fluctuations, which vanish at zero temperature, and reduce the stiffness at finite temperatures.

We compute Eq. (7) by a Monte Carlo (MC) simulation of $H_{3dXY}$ with the Wolff cluster updates algorithm [27], see Appendix D for details. We choose $L_c = L_b = 60$, and vary the width in the range $L_a \in \{40, 50, \ldots, 80\}$ using the anisotropy parameters in the range $\alpha = 0.01 - 0.02$. The minimal accessible anistropy parameter is determined by the maximal lattice size.

In Fig. 4, we plot the intra-plane stiffness $\rho^\parallel$ as a function of temperature $T$, and width $L_a$. The ansitropy parameter is fixed at $\alpha = 0.01$. $T_c \simeq 1.086$. The expected thermodynamic critical behavior, Eq. (6), is depicted by a dashed line in Fig. 4. For the disorder-free model, the tail above $T_c$ indicates that the in-plane correlation length exceeds $L_a$. Thus, a larger $L_a$ reduces the width of the tail. For millimeter scale superconducting rings, this tail should be unobservably small.

In Fig. 5, the MC data for $\rho^\perp(T)$ are shown as points. Given a numerical resolution threshold $\varepsilon$, $\rho^\perp(T)$ appears to vanish at transition temperatures $\widetilde{T}_c^\perp$ which depend on $\varepsilon$ and the width $L_a$. The solid lines and the inset describe a fit of the MC data to analytic formulas derived in the following Section.

## 5  Crossover to one dimensional Josephson array

The *apparent* premature vanishing of $\rho^\perp$ in a finite size sample of an approximately unit aspect ratio, is due to its crossover to a quasi one-dimensional behavior as $T \to T_c$. The stiffness of a one dimensional (1d) classical XY chain with inter-site coupling $J_{1d}$, lattice constant $a$ and chain length $L$ is well known,

$$
\rho_{1d}(T, L) = TLZ_2/Z_0
$$
$$
Z_{2p} = \sum_{n=-\infty}^{\infty} \left( \frac{I_n(J_{1d}/T)}{I_0(J_{1d}/T)} \right)^{L/a} n^{2p},
$$
(8)

where $I_n$ are modified Bessel functions and $p = 0, 1$. Luttinger liquid (LL) theory [8, 28], which applies at $L \gg a$, yields an analytic result where $\rho_{1d}$ depends on the dimensionless variable $x \equiv LT/(J_{1d}\,a)$ as,

$$
\rho_{\mathrm{LL}}(J_{1d}, x) = J_{1d}\,a \left( 1 - \frac{\pi^2}{x} \frac{\vartheta_3''(0, e^{-2\pi^2/x})}{\vartheta_3(0, e^{-2\pi^2/x})} \right)
$$
$$
\simeq J_{1d}\,a \begin{cases} 1 & (x \le 2) \\ 20 \exp(-0.472\,x) & (x \ge 10) \end{cases} \quad,
$$
(9)

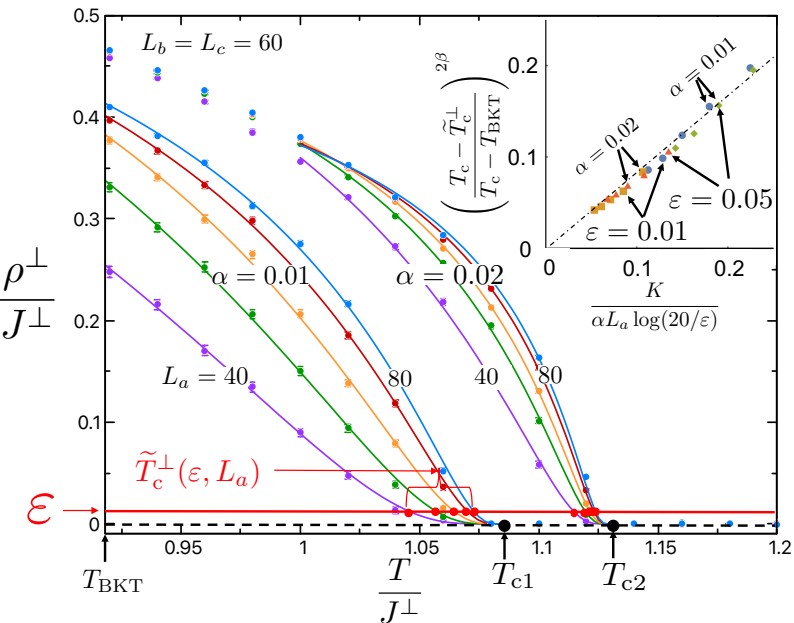

Figure 5: MC evaluations of $\rho^\perp$ for the clean 3dXY model Eq. (2), a function of temperature for a range of sample widths $L_a \in \{40, 50, \ldots, 80\}$, and anisotropy parameters $\alpha$. The thermodynamic critical temperatures are evaluated as $T_{c1} = 1.086 J_a$ and $T_{c2} = 1.13 J_a$ for $\alpha = 0.01$ and $0.02$, respectively. Solid lines are best fits to Eq. (12). $\varepsilon$ is arbitrarily chosen as the numerical resolution which defines the apparent transition temperatures $\widetilde{T}_c^\perp$ by Eq. (14). Inset: Verification of Eq. (15) by collapse of all the temperature shifts for various $L_a, \varepsilon, \alpha$ obtained from the main graphs.

where $\vartheta_3(z, q) = 1 + 2\sum_{n=1}^\infty q^{n^2} \cos(2nz)$, and prime denotes differentiation with respect to $z$. Comparison between Eqs. (8) and (9) is shown in Appendix B.

Now we return to the $c$-axis stiffness $\rho^\perp$ of the layered model (2), which can be described as a chain of Josephson junctions along the $c$-axis with inter-grain coupling,

$$J_{\text{eff}}(T) = \frac{L_a L_b}{(\xi^\parallel)^2} \times J^\perp \Delta^2(T), \tag{10}$$

Toward $T_c$, $\Delta^2(T)$ vanishes as $t^{2\beta}$ by Eq. (5). Substituting $J_{1d} = J_{\text{eff}}(T)$ we expect the asymptotic behavior of Eq. (9) to be realized after replacing

$$x \to \frac{L_c T}{J_{\text{eff}}(T)\, \xi^\perp}. \tag{11}$$

Thus for $t \ll 1$, $x \gg 1$ and

$$\rho^\perp(T) \approx 20\, \rho^\perp(T_{\text{BKT}}) \exp\left(-\frac{K}{\alpha L_a}\, t^{-2\beta}\right), \tag{12}$$

$$K \simeq \frac{0.472\, r T_c\, (\xi^\parallel)^2}{J_a\, \Delta_{\text{BKT}}^2\, \xi^\perp}. \tag{13}$$

For any experimental resolution $\varepsilon$, an apparent vanishing temperature $\widetilde{T}_c^\perp(\varepsilon)$ is defined by the threshold condition,

$$\frac{\rho^\perp(\widetilde{T}_c^\perp)}{\rho^\perp(T_{\text{BKT}})} = \varepsilon. \tag{14}$$

By Eq. (12), the apparent width dependent transition temperature is,

$$T_c - \widetilde{T}_c^\perp = (T_c - T_{\text{BKT}}) \left( \frac{K}{\alpha L_a \log(20/\varepsilon)} \right)^{1/2\beta} \tag{15}$$

The most important consequence of the quasi one-dimensional behavior, is that the temperature shifts are proportional to $(\alpha L_a)^{-1/2\beta}$. This is a much larger shift than expected from critical fluctuations, which are of order $(\alpha L_a^2)^{-1}$.

In the inset of Fig. 5 we verify the validity of Eq. (15) by collapsing of all the temperature shifts onto a straight line. The slope of this line differs only by 20% from unity, which we attribute to the choice of the (non-universal) constants in the asymptotic expression of Eq. (9).

## 6   Comparison of Theory to Experiments

In comparing Eq. 15 to the MC results, we have used the 3dXY critical exponent $\beta = 0.349$.

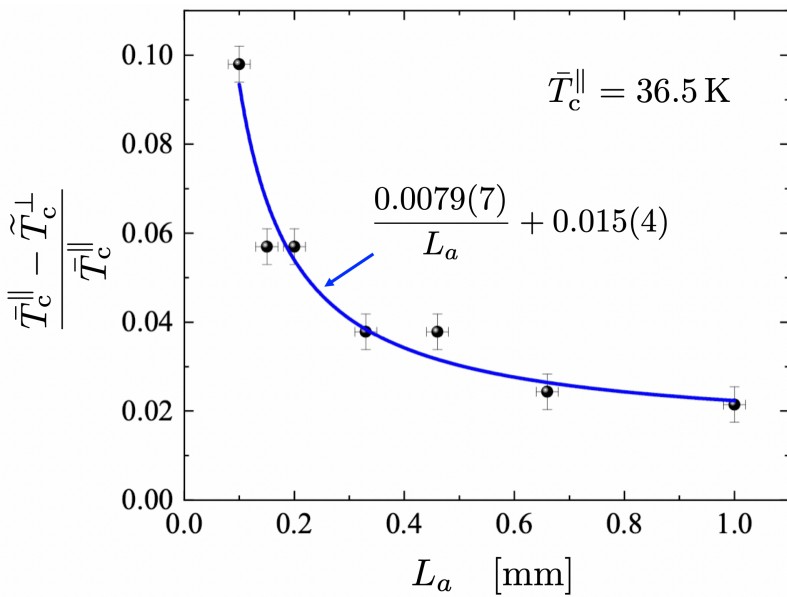

Figure 6: Comparison of experimental results for $La_{1.83}Sr_{0.17}CuO_4$ rings of Fig. 2 and theoretical prediction of Eqs. (15) and (19). Crosses: The apparent $c$-axis transition temperature shifts $\widetilde{T}_c^\perp$ of the $a$-rings, as determined in Fig. 2(d). $L_a$ are the bottleneck widths of the $ab$ planes. Line: the least square fit using $\alpha^{\text{fit}} = 4.1 \times 10^{-5}$. The offset of the reduced temperature 0.015(4) agrees with the estimated layer-correlated disorder (see text). We use the mean field exponent $\beta = \frac{1}{2}$, due to the narrow Ginzburg critical regime near $T_c$.

For the experimental $La_{1.83}Sr_{0.17}CuO_4$ crystals, the millimeter width corresponds to $\sim 10^5$ effective lattice constants, and the anistropy parameter will turn out to be $\alpha < 10^{-4}$, which yields an unobservable narrow Ginsburg critical region. Hence we shall fit the Eq. 15 with the mean field exponent $\beta = 0.5$.

The $a$-ring is sequentially cut such that the induced current is governed by the bottleneck region. There, the induced current flows along the $c$ axis over an effective length

of $L_c^{\text{eff}} = 2$ mm. The transverse dimension $L_b = 0.46$ mm yields $r = 4.34$. The width is varied in the range $L_a \in [0.1, 1]$ mm. We estimate the experimental resolution at $\varepsilon = 10^{-2}$. (High accuracy of $\varepsilon$ is not essential, since $\widetilde{T}_c^\perp$ depends on it logarithmically).

At zero temperature the coherence lengths have been determined experimentally [29] to be $\xi^\parallel(0) \simeq 3$ nm and $\xi^\perp(0) \simeq 1.3$ nm. The in-plane lattice constant for the effective 3dXY model is the coherence length estimated at $T_{\text{BKT}}$ to be $\xi^\parallel(T_{\text{BKT}}) = \xi^\parallel(0)/\Delta_{\text{BKT}}$. Due to the incoherent single-electron tunneling between the layers, we assume that the Cooper pair size in the $c$ direction remains confined to a single plane $\xi^\perp(T_{\text{BKT}}) \simeq \xi^\perp(0)$.

In Fig. 6, the apparent $c$-axis transition temperatures $\widetilde{T}_c^\perp(L_a)$ are plotted. The data is somewhat noisy, presumably because of the introduction of deep cuts during the ring's cutting process, which are eliminated by subsequent cuts. The two-parameter fit function is plotted,

$$\frac{\bar{T}_c^\parallel - \widetilde{T}_c^\perp}{\bar{T}_c^\parallel} = \frac{A}{L_a[\text{mm}]} + (\delta t)_{\text{dis}} \tag{16}$$

with $A = 0.0079$ and $(\delta t)_{\text{dis}} = 0.015(4)$. The dimensionless temperature shift $(\delta t)_{\text{dis}}$ is understood as the effect of layer-correlated inhomogeneity (see Appendix A). We use the high temperature tail of magnitude $(\delta T_c^\parallel)_{\text{dis}} \simeq 0.5$ K, which is depicted in Fig. 2(e). Subtracting $(\delta T_c^\parallel)_{\text{dis}}$ from $\bar{T}_c^\parallel = 36.5$ K yields a bound for $\widetilde{T}_c^\perp$ for wide samples,

$$\lim_{L_a \to \infty} \widetilde{T}_c^\perp = \bar{T}_c^\parallel - (\delta T_c^\parallel)_{\text{dis}} = 36 \text{ K}. \tag{17}$$

The estimated layer-correlated disorder shift is consistent with the fit in Fig. 6,

$$(\delta t)_{\text{dis}} \equiv \frac{(\delta T_c^\parallel)_{\text{dis}}}{T_c^\parallel} \in 0.015(4). \tag{18}$$

Using Eqs. (3), (4), (13) and (15), and the parameters listed above we obtain

$$A = \Delta T_c(\alpha) \frac{0.472 \times 10^{-6} \, r \, T_c \, (\xi^\parallel)^2}{\alpha \, \Delta_{\text{BKT}}^4(\alpha) \, \xi^\perp \log(20/\varepsilon)} = 0.0079 \tag{19}$$

which can be fit by the anisotropy parameter,

$$\alpha^{\text{fit}}(T \simeq T_c) = 4.1 \times 10^{-5} \tag{20}$$

# 7 Discussion and Summary

The experimental conundrum, which was first noted in Ref. [3], was that stiffness measurements of $a$-rings and $c$-rings, cut out from same cuprate crystal, exhibited different transition temperatures. In this paper, we have shown that this difference cannot be fully explained by layer-correlated disorder, since it varies consistently with the layer's width, which is not coupled to the distribution of layer-correlated disorder.

With the help of Monte-Carlo simulations, inter-layer mean field theory, we have identified a narrow regime below the bulk transition temperature $T_c$ where the inter-layer stiffness of finite size samples crosses over to an effective one dimensional Josephson array behavior. As a result, we resolve the conundrum, and explain the Monte-Carlo data, as a width-dependent, *apparent* reduction of the $c$-axis $T_c$. The visibility of the effect depends on the smallness of the anisotropy parameter $\alpha$.

We note that $\alpha^{\text{fit}}$ parametrizes the effective Hamiltonian near $T_c$. We compare it to the zero temperature anisotropy parameter reported for optimally doped $\text{La}_{2-x}\text{Sr}_x\text{CuO}_4$ (for $\text{Sr}_{0.15}$) in Ref. [30],

$$\alpha^\lambda(T = 0) = \left(\frac{\lambda_c}{\lambda_a}\right)^{-2} = 4.6 \times 10^{-3}. \tag{21}$$

The difference in anisotropy can be attributed to the reduction of inter-plane coherence due to thermally excited nodal quasiparticles of the $d$-wave superconductor and the effects of inter-planar vortex rings above the two dimensional $T_{\text{BKT}}$.

*Analog in* $^4He$ – We have seen that $\alpha \ll 1$ can be mapped onto an isotropic model on samples with large aspect ratio. A similar "premature" vanishing of $\rho^\perp$ has been observed on a quasi-one dimensional brick, *i.e.* $L_a \ll L_c$ [7]. This result was used to explain the experimental disappearance of superfluid density of $^4$He embedded in quasi one-dimensional nanopores [31, 32]. Here we explain the *apparent* reduction of $\widetilde{T}_c(L_a)$, not as a true thermodynamic transition but rather as a consequence of an essential singularity decay of $\rho^\perp$ toward the thermodynamic $T_c$.

In general, layered superconductors with very high anisotropy are expected to exhibit such apparent differences between transition temperatures of in-plane and out of plane persistent currents. For example, an emergent anisotropy of layered superconductors has been an important consequence of certain pair density wave (PDW) ordering [33]. We propose that the dependence of inter-layer stiffness transition temperatures on sample width cold help characterize the emergent anisotropy parameter of that interesting PDW phase.

# 8 Acknowledgements

A.S. and I.M. contributed equally to this work. We thank Erez Berg, Snir Gazit, Dror Orgad, and Daniel Podolsky for beneficial discussions. A.A. acknowledges the Israel Science Foundation (ISF) Grant No. 2081/20. A.K. acknowledges the ISF Grant. No. 1251/19 and 3875/21. This work was performed in part at the Aspen Center for Physics, which is supported by National Science Foundation grant PHY-2210452, and at the Kavli Institute for Theoretical Physics, supported by Grant Nos. NSF PHY-1748958, NSF PHY-1748958 and NSF PHY-2309135.

# A  Planar correlated disorder

Refs. [5, 6] have considered the layered $XY$ model where the $ab$ planes exhibit a variable $z$-dependent stiffness $\rho^\|(z)$ for $z \in [0, L_c]$. We can see the effects of bounded correlated disorder on superconductors with a variation of $\rho^\|(z)$ along the $c$-axis. In each segment, the stiffness temperature dependence has a different $T_c$,

$$\rho^\|(T) = \rho^\|(0) \left|\frac{T - T_c^\|(z)}{\bar{T}_c^\|}\right|^{2\beta - \eta\nu} \tag{22}$$

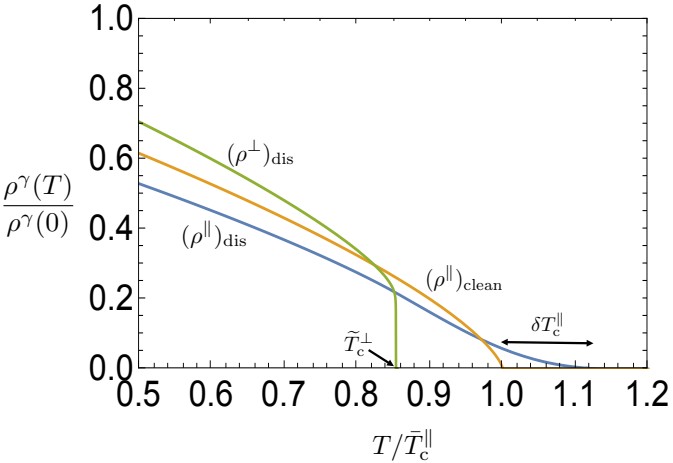

Figure 7: Effects of planar-correlated disordered, modelled by $c$-axis dependent in-plane stiffness $\rho^\parallel(z)$, with an average transition temperature $\bar{T}_{\mathrm{c}}^\parallel$ and a width of transition temperatures $(\delta T_{\mathrm{c}}^\parallel)_{\mathrm{dis}} = 0.1\bar{T}_{\mathrm{c}}^\parallel$. Orange line: The clean system with a three dimensional critical behavior. Blue line: the global $\rho^\parallel$ showing a disorder induced high temperature tail above $\bar{T}_{\mathrm{c}}^\parallel$. Green line: the global $\rho_\perp^{\mathrm{dis}}$, which is dominated by the weakest interplane stiffnesses, and vanishes below $T_{\mathrm{c}}^\parallel$.

where $T_{\mathrm{c}}^\parallel(z) \propto \rho^\parallel(z,0)$ is the local transition temperature whose average is defined as $T_{\mathrm{c}}^\parallel$ and maximal variation is $(\delta T_{\mathrm{c}}^\parallel)_{\mathrm{dis}}$. The global $ab$-stiffness is given by the integral

$$\rho^\parallel = \rho^\parallel(0) \int\limits_0^{L_c} \frac{dz}{L_c} \left| \frac{T - T_{\mathrm{c}}^\parallel(z)}{\bar{T}_{\mathrm{c}}^\parallel} \right|^{2\beta - \eta\mu} \quad , \tag{23}$$

which smears the average critical temperature $\bar{T}_{\mathrm{c}}^\parallel$ by a high temperature tail at $T \in \left[ \bar{T}_{\mathrm{c}}^\parallel, \bar{T}_{\mathrm{c}}^\parallel + (\delta T_{\mathrm{c}}^\parallel)_{\mathrm{dis}} \right]$.

In contrast, the $c$-axis stiffness $\rho^\perp(z)/\rho^\perp(0)$ is proportional to the local order parameter squared $\Delta(z) \propto |T - T_{\mathrm{c}}(z)|^\beta$. The global $c$-axis stiffness is the harmonic average given by,

$$\rho^\perp = \rho^\perp(0) \left( \int\limits_0^{L_c} \frac{dz}{L_c} \frac{(\bar{T}_{\mathrm{c}}^\parallel)^{2\beta}}{|T - T_{\mathrm{c}}^\parallel(z)|^{2\beta}} \right)^{-1} \quad . \tag{24}$$

The weakest segment, with the lowest $\rho^\perp(z)$, dominates the integral. The temperatures where the order parameter of this segment vanishes is

$$\widetilde{T}_{\mathrm{c}}^\perp \leq \bar{T}_{\mathrm{c}}^\parallel - (\delta T_{\mathrm{c}}^\parallel)_{\mathrm{dis}} \quad , \tag{25}$$

above which the global $\rho^\perp(T)$ disappears. The effect of bounded layer-correlated disorder is demonstrated in Fig. 7.

## B  Asymptotic behavior of stiffness of a one dimensional $XY$ chain

In Fig. 8 we depict the exact result of the stiffness of the one dimensional XY chain as given by Eq. (7) of the main text. At large $L/a$ the graphs show convergence to the

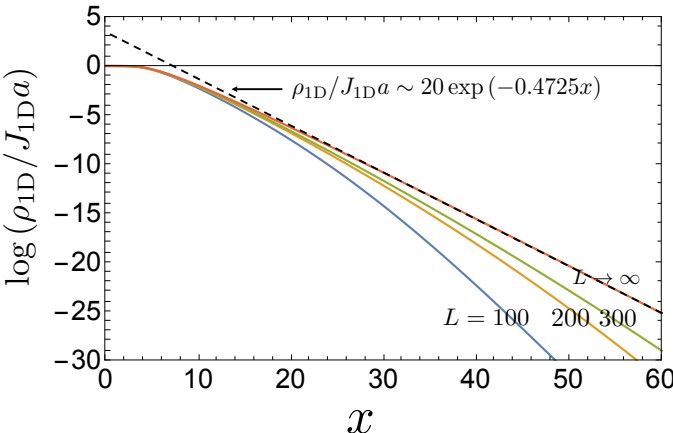

Figure 8: Stiffness as a function of scaled variable $x = LT/(J_{1d}a)$ in the one dimensional XY model for different lengths $L$ as given by the exact result of Eq. (7), and asymptotically at $L \to \infty$ by Eq. (8) of the main text.

analytic Luttinger-Liquid form [8, 28], which at large $x$ is given by Eq. (8) of the main text.

## C  Estimation of finite size shift in $T_c$

Fine size scaling produce unobservably small finite size shifts of $T_c$ for millimeter size samples, as shown by the following. The correlation lengths above $T_{BKT}$ diverge as

$$\xi^{\parallel}(t) \simeq \frac{1}{\sqrt{\alpha}} \xi_a^{(0)} t^{-\nu} \quad , \quad \xi^{\perp}(t) \simeq \xi_c^{(0)} t^{-\nu} \tag{26}$$

where we use the Ginzburg-Landau definition of correlation lengths, $\xi_\gamma^{-1} \propto \sqrt{\rho_\gamma}$, to obtain the factor of $\sqrt{\alpha}$ between the divergent correlation length.

For Eq. (2) with sample dimensions $L_\gamma, \gamma = a, b, c$ the stiffness components near $T_c$ vanish as [34],

$$\frac{\rho_c}{\rho_c(T_{BKT})} = t^v \, \Phi[x_a] \quad , \quad x_a = \xi_a(t)/L_a. \tag{27}$$

where $\Phi(x)$ is differentiable function with a finite value at $x = 0$. We expand $\Phi$ to linear order in $\xi_a$ and set $\rho^{\perp} \to 0$ to obtain,

$$0 = \Phi_0 + \partial_{x_a}\Phi \times \left( \frac{t^{-\nu}\xi_a^{(0)}}{\sqrt{\alpha}L_a} \right) + \mathcal{O}(x_a^2) \tag{28}$$

which is solved by a positive shift of $T_c$ by the amount

$$\delta t \simeq -\frac{\Phi_0}{\partial_a \Phi_\gamma} \left( \frac{\sqrt{\alpha}L_a}{\xi_a^{(0)}} \right)^{-1/\nu} . \tag{29}$$

For the experimental $La_{1.83}Sr_{0.17}CuO_4$ rings, taking $\alpha = 10^{-5}$, $L_a/\xi_a^{(0)} \sim 10^6$, yields $|\delta t| \le 10^{-4}$, which is much below experimental temperature resolution.

# D    Details of the Monte-Carlo simulations using cluster algorithm

The superfluid stiffness or the helicity modulus (with $a_\gamma = 1$) for the classical Hamiltonian of Eq. (2) is given by Eq. (7) [7, 26, 35].

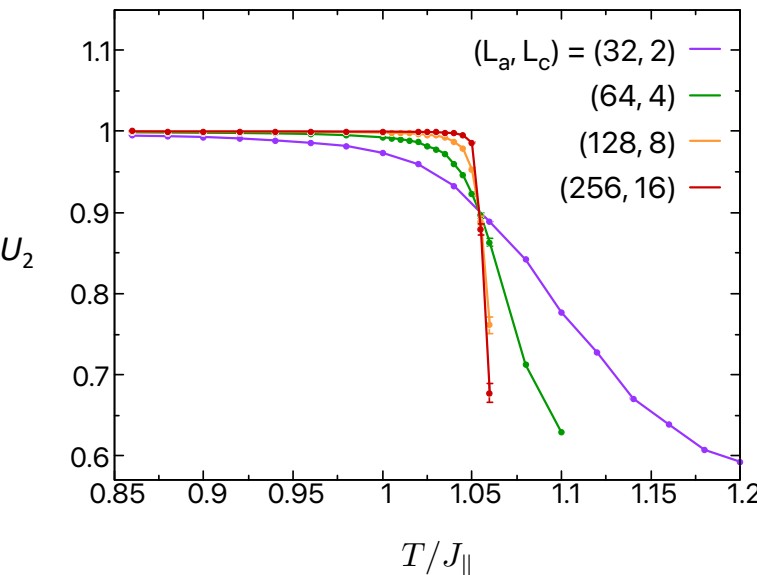

Figure 9: Binder cumulant $U_2$ plotted as a function of temperature $T$, for anisotropy $\alpha = 0.005$ and for different sizes with a fixed aspect ratio $L_a/L_c = 16$ and $L_b = L_a$.

In the Wolff-cluster algorithm [27], we assume the XY spins $\boldsymbol{S}$ to be the unit vectors in $\mathbb{R}^3$. In every Monte-Carlo (MC) step, we first choose a random site $\boldsymbol{r} \in \mathbb{R}_3$ and a random direction $\boldsymbol{d} \in S_2$, and consider a reflection of the spin on that site about the hyperplane orthogonal to $\boldsymbol{d}$. Note that this is equivalent to the spin-flipping operation in Ising model. We then travel to all neighboring sites ($\boldsymbol{r}'$) of $\boldsymbol{r}$, and check if the bond $\langle \boldsymbol{r}\boldsymbol{r}' \rangle$ is activated with a probability

$$P_\gamma(\boldsymbol{r}, \boldsymbol{r}') = 1 - \exp\Big( \min\big[ 0, 2 J_\gamma \beta (\boldsymbol{d} \cdot \boldsymbol{S_r})(\boldsymbol{d} \cdot \boldsymbol{S_{r'}}) \big] \Big), \tag{30}$$

where $\beta$ is the inverse temperature. If this satisfies, we mark $\boldsymbol{r}'$ and include it to a cluster $\mathcal{C}$ of "flipped" spins. We iteratively continue this process for all unmarked neighboring sites of $\boldsymbol{r}'$ and grow the cluster size until all the neighbors turn out to be marked. We use such $10^6$ number of MC steps for thermalization, followed by another $10^7$ number of MC steps for measurement of different observables, such as the helicity modulus and the binder cumulant. We estimate the errors of different observables by using a standard Jackknife analysis of the MC data.

In Fig. 3 of the main text, we have presented the inter-plane superfluid stiffness $\rho^\perp$ for different system sizes of $L_a \in [60 - 80], L_b = L_c = 60$, near the transition.

Next we calculate the Binder cumulant, defined in terms of the higher powers of magnetization $m$ as following [36]

$$U_2 = \frac{3}{2} \left( 1 - \frac{1}{3} \frac{\langle m^4 \rangle}{\langle m^2 \rangle^2} \right), \tag{31}$$

and we use it to extract the value of critical temperatures accurately. As an example, in Fig. 9, we present $U_2$ as a function of $T$, for an anisotropy parameter $\alpha = 0.005$ and for

different system sizes with a fixed aspect ratio $L_a = L_b$ and $L_a/L_c = 16$. In the ordered phase when all the spins are aligned it takes a value 1, while in the disordered phase it vanishes and takes an intermediate value between 0 and 1 at the critical point. Therefore, by tracking the crossing between different system sizes, we find a critical temperature $T_c \sim 1.05$ for these parameters. using a similar analysis, we obtain $T_c$ for other anisotropy parameters also, discussed in the main text.

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
