# Peer review of "The two critical temperatures conundrum in La$_{1.83}$Sr$_{0.17}$CuO$_4$"

_SciPost Physics_

## Round 4 · Referee Report · Anonymous (Referee 1) · 2024-3-10

Strengths

  1. Innovative experimental method;
  2. Very deep theoretical analysis;
  3. Well-organised paper.

Weaknesses

  1. The experiments were performed on a small number of samples.

Report

The paper reports experimental measurements of the anisotropic superconducting stiffness of La1.83Sr0.17CuO4 using the original experimental method. The authors observe an unexpectedly strong effect of the sample size on the stiffness close to the superconducting transition temperature. This effect is explained by the crossover to the 1D regime close to Tc. Using Monte Carlo simulations, the anisotropy ratio of lambda_c/lambda_ab is obtained close to Tc, which is much larger than zero temperature anisotropy. The paper is interesting, well-written and deserves publication.

Requested changes

Authors may optionally consider the following questions:
Can the authors exclude oxygen content variation during the ring-cutting process, especially at small ring sizes? How reproducible are the data shown in Fig.2 e,d? According to the authors, only two rings survived the size reduction procedure. Is there any valuable information from broken rings? Can the authors show other data?

The authors mentioned, "We note that the c-ring magnetizations exhibit a high-temperature tail of ≃ 0.5 K above the extrapolated transition at T¯c∥. This is attributed to layer-correlated inhomogeneity, as discussed in the Introduction and Appendix A." Why do inhomogeneities play a role only for the c-ring?

  • validity: good
  • significance: high
  • originality: high
  • clarity: high
  • formatting: excellent
  • grammar: excellent

Author:  Assa Auerbach  on 2024-04-16  [id 4421]

(in reply to Report 1 on 2024-03-10)
Category:
answer to question

We Thank the Referee for the thoughtful Report and constructive questions. The Revised manuscript incorportaed the answers into the Introduction and Summary sections. The questions are answered in detail below.

Referee > Can the authors exclude oxygen content variation during the ring-cutting process, especially at small ring sizes?
Reply> We thank the Referee for raising this question.
We have applied exactly the same laser cutting or polishing process to both c-rings and a-rings. As shown in Figure 1, except for the narrowest c-ring sample (of dimensions 0.19/0.09), which we suspect has suffered a significant structural damage, all the other samples show very much the same Tc’s, and therefore no effect of the cutting and polishing process. If significant oxygen content was varied during the cutting process, the c-ring transition temperatures Tc^parallel would not stay constant as we cut and polished them.

In the a-rings, there is a definite systematic reduction in the apparent transition temperature, which we therefore associate with a finite size effect.

Referee> How reproducible are the data shown in Fig.2 e,d? According to the authors, only two rings survived the size reduction procedure. Is there any valuable information from broken rings? Can the authors show other data?

Reply> The data was reported for the a and c rings which survived the full sequential cutting procedure. The data for each width of the rings was totally reproducible. Broken rings bring no information to this experiment, since they do not close the persistent current loop, and have zero magnetization signal. The systematic finite size dependence of Tcperp is our main experimental discovery, which brought to light an interesting theoretical puzzle.
The experimental effort producing the data in Fig.2 e,d, was a sizeable fraction of a Ph.D. thesis: crystal growth, and sequences of cutting, polishing and measurements. We encourage other groups to perform similar measurements and find out if they obtain different results.
Referee> The authors mentioned, "We note that the c-ring magnetizations exhibit a high-temperature tail of ≃ 0.5 K above the extrapolated transition at Tcparallel. This is attributed to layer-correlated inhomogeneity, as discussed in the Introduction and Appendix A." Why do inhomogeneities play a role only for the c-ring?

Reply> Layer-correlated inhomogeneity affects the stiffness of both c and a rings, but in a different way, as explained in Appendix A and Figure 7.
The c-ring diamagnetization (whose persistent currents flow within the layers) is determined by the layer averaged in-plane stiffness, and therefore vanishes above the layer-averaged <Tcparallel>. The width of the high temperature tail indicates the width of the above-average in-plane stiffness distribution.
The a-ring’s magnetization is governed by layers with the smallest order parameters. Therefore it vanishes when the order parameters in these layers disappear, which happens at a lower temperature than <Tcparallel>. Therefore layer-correlated disorder can explain a difference between a and c ring Tc’s.
However, as discussed in the paper, for the magnitude of this shift, and more importantly, its functional dependence on the width La , the layer-correlated disorder is insufficient.

---

## Round 4 · Referee Report · Anonymous (Referee 2) · 2024-3-19

Strengths

1-simple yet effective experimental setup
2-in-depth modeling of obtained data according to established theory

Weaknesses

1-readability of paper

Report

The paper contains apparently straightforward experiments that speak for themselves. This is already a good reason to publish the paper. In addition, a theoretical analysis is given that explains the findings in terms of geometric effects, other possible origins for the apparently different transition temperatures are discussed. It appears that one can treat the superconducting state of this material according to their assumptions, while there have been reports on different temperatures involved in the transition years ago (Haase et al. J. Phys.: Condens. Matter 21, 455702). Likely, such microscopic details may not show up in these more macroscopic finite size effects.
Therefore, the paper should be published.

Requested changes

The paper is somewhat difficult to read. It has a rather simple, easy to convey experimental part that meets a lengthy theoretical treatment. The overall structure of the paper is somehow, at least, unusual. The Discussion and Summary is shorter than any other part of the paper, that has at least two striking typos 'crosover' and 'cold help'. There are more typos, even if one does not count those in the supplement. Here, in their own interest, the authors should use regular software to check on things.

  • validity: high
  • significance: high
  • originality: high
  • clarity: ok
  • formatting: acceptable
  • grammar: reasonable

Author:  Assa Auerbach  on 2024-04-16  [id 4422]

(in reply to Report 2 on 2024-03-19)

We thank the Referee for the constructive Report . The proposal to improve the discussions is well taken. We have followed the recommendation and extended the Introduction and Summary and Discussion Sections (1 and 7) in the resubmitted manuscript, which clarified the theoretical message. We have eliminated typos. We believe that the readability of the paper has been improved.

---

## Round 5 · Referee Report · Anonymous (Referee 2) · 2024-4-18

Report

The current version could be published, but there is a new, subtle problem with sample notation. In title and abstract it is clear that this paper is only on a particular doping, while the authors go and denote this sample, probably out of convenience, with LSCO, which denotes the whole family. Now, later in the text LSCO is also used for other doping levels. So I suggest that the authors introduce a special name for their sample if they do not want to write out the longer version La_{1-x}Sr_xCuO_4.

Recommendation

Ask for minor revision

  • validity: -
  • significance: -
  • originality: -
  • clarity: -
  • formatting: -
  • grammar: -

Author:  Assa Auerbach  on 2024-04-18  [id 4429]

(in reply to Report 1 on 2024-04-18)

Thank you for alerting us to the problematic acronym LSCO.
We will make sure to return to the longer notation : La_{1-x}Sr_xCuO_4 for various values of x, and distinguish between samples of different Sr concentrations.

---

## Round 5 · Referee Report · Anonymous (Referee 1) · 2024-4-28

Report

I am generally satisfied with the authors' replies to my questions.
The only concern I have is how the authors present the details of the sample preparation and characterisation. In particular, could the authors provide information (or a reference) on the growth temperature and time and how doping was determined? Also, throughout the paper, the same short name of the sample, "LSCO", is used to refer to the samples with different doping levels as well as to the system as a whole. To avoid possible confusion, I would recommend using a bit of an extended sample name for a specific sample, for example, LSCO_0.17.

Recommendation

Ask for minor revision

---

## Editorial Decision

resubmitted